# Ecological and biogeographic drivers of biodiversity cannot be resolved using clade age-richness data

Daniel L. Rabosky [1,2 ✉] & Roger B. J. Benson[3]

Estimates of evolutionary diversification rates – speciation and extinction – have been used extensively to explain global biodiversity patterns. Many studies have analyzed diversification rates derived from just two pieces of information: a clade's age and its extant species richness. This "age-richness rate" (ARR) estimator provides a convenient shortcut for comparative studies, but makes strong assumptions about the dynamics of species richness through time. Here we demonstrate that use of the ARR estimator in comparative studies is problematic on both theoretical and empirical grounds. We prove mathematically that ARR estimates are non-identifiable: there is no information in the data for a single clade that can distinguish a process with positive net diversification from one where net diversification is zero. Using paleontological time series, we demonstrate that the ARR estimator has no predictive ability for real datasets. These pathologies arise because the ARR inference procedure yields "point estimates" that have been computed under a saturated statistical model with zero degrees of freedom. Although ARR estimates remain useful in some contexts, they should be avoided for comparative studies of diversification and species richness.

[1] Museum of Zoology, University of Michigan, Ann Arbor, MI, USA. [2] Department of Ecology and Evolutionary Biology, University of Michigan, Ann Arbor, MI 48109, USA. [3] Department of Earth Sciences, University of Oxford, Oxford OX1 3AN, UK. ✉email: drabosky@umich.edu

The causes of large-scale variation in species richness—in time, in space, and among clades—remain poorly understood. However, it is increasingly clear that much of this variation results from differences in evolutionary rates of speciation and extinction[1–4], and determining the drivers of these rate differences is a major goal of macroevolutionary and macroecological research. Consequently, there is widespread interest in developing methods for quantifying speciation and extinction rates, from molecular phylogenies and the fossil record[5–9]. One of the most widely-used methods for studying these rates is also the simplest. This approach uses the age of a clade and its present-day species richness to compute a point estimate of the net rate of species diversification through time, which is simply the difference between the rate at which new species are gained through speciation ($\lambda$) and the rate at which they are lost by extinction ($\mu$). This net rate of diversification ($r = \lambda - \mu$) is a key parameter for stochastic models of species diversification and can be used to predict the mean and variance of clade diversity through time[10]. For the constant-rate birth-death process that begins with a single ancestral species, we can easily compute an estimate of the net diversification rate as

$$r = \frac{1}{t} \log[n(1 - \varepsilon) + \varepsilon] \qquad (1)$$

where $\varepsilon$ is the extinction fraction ($\mu/\lambda$) and $t$ is the elapsed time from the start of the process. This point estimate is the maximum likelihood estimator for the process (derivation in Supplementary Note 1) and is sometimes labeled the method-of-moments estimator or the Magallon-Sanderson estimator[11]. A related process with nearly identical mathematical properties can be applied to crown clades (e.g., two ancestral species). Here, we refer to the use of Eq. (1) and other variants of net diversification rate for a single clade as "Age-Richness-Rate" (ARR) estimators, to emphasize the three quantities involved (clade age, species richness, diversification rate). Once researchers have two numbers (an estimate of a clade's age and its richness), the ARR method discards any further information in favor of a simple summary statistic (Eq. (1)), similar to an arithmetic mean.

The simplicity of the ARR estimator has led to its widespread use in comparative studies of species richness, many of which attempt to explain variation in the global or regional biodiversity of groups through the analysis of clade-specific ARR estimates[12–16]. Other studies test whether specific organismal traits and biogeographic features are correlated with ARR values, as a step toward understanding why some clades have more species than others[17–22]. ARR estimators have thus become a convenient shortcut to inferring the drivers of variation in species richness when no virtually no information from the fossil record or molecular phylogenies is available. This shortcut comes with strong assumptions: ARR estimators assume that the present-day richness of a clade was generated under a constant, positive net diversification rate, with invariant speciation and extinction rates through time. While most researchers would recognize that evolutionary rates have not been constant through time, there is a general perception that the simplicity of the method implies robustness to violation of its assumptions.

ARR estimators can be viewed as a model-based estimate, but where the inference model is fully saturated. A single bivariate datum (age, richness) is used to estimate a single parameter ($r$), conditional on a particular value of $\varepsilon$ that is chosen in advance by the researcher. Because the ARR model is saturated, the age and diversity numbers used to compute the ARR estimate cannot be used to test the adequacy of the inference model, and the ARR estimate for a given age-richness datum will always perfectly predict the input data with zero error (Supplementary Fig. 1). Assessing model adequacy, therefore, requires information from additional clades, from the fossil record, or from time-calibrated phylogenetic trees.

Here, we provide a comprehensive assessment of ARR estimators from a theoretical and empirical perspective. We first ask whether the ARR estimators describe an "identifiable" process by performing a mathematical comparison between a scenario with positive net diversification ($r > 0$) and one where the net diversification rate is exactly zero ($r = 0$). We then determine whether there is enough information in cross-clade comparative datasets to justify the assumption that net diversification rates differ across clades.

For our empirical assessment, we test both the stability and predictive accuracy of ARR estimates using paleontological time series of species richness. If rates from ARR estimators can be compared among clades that vary widely in age, then the value of the estimator should not depend strongly on the time of observation within the history of a single clade. In this context, the species richness of each clade observed during the present-day can be regarded as representing just an arbitrarily chosen moment in its history. We then ask whether a given ARR estimate for a clade can be used to predict the species richness of the same clade at another time point in its history (Fig. 1). This comparison represents a critical test of ARR adequacy and can only be performed by incorporating paleontological information. ARR estimates represent a "saturated" model, so there is no information remaining in a typical comparative study (of present-day species richness) with which to test model fit, because the computed ARR rate will always predict perfectly the observed richness for a given clade. However, if ARR estimates index a biologically-meaningful process, they should at least retain some ability to predict species richness at other points in a clade's history. We compare the predictive ability of ARR estimators to nonbiological alternatives, including a simple scenario where diversity fluctuates randomly around a single value ("constant" scenario), as well as a formal birth–death model specifying a net diversification rate of zero. By assessing the predictive accuracy of ARR estimates with respect to paleontological time-series data, we provide a first test of the assumption that ARR estimates describe a macroevolutionary property of clades that can be explained by biological or biogeographic traits. Our approach, therefore, tests the adequacy of ARR estimates for use in cross-clade comparative studies.

## Results

**Mathematical analysis.** In Supplementary Note 1, we prove that the probability of a given age-richness datum under the ARR estimator is identical across the extinction fraction domain ($0 \leq \varepsilon \leq 1$). For any value of net diversification greater than or equal to zero, the probability of the observed data (species richness, $n$; and clade age, $t$) can be shown to equal

$$P_{n,t|\varepsilon} = \left(\frac{1}{n}\right)\left(\frac{n-1}{n}\right)^{n-1} \qquad (2)$$

and is therefore independent of both clade age and the extinction fraction. There is no information in the data that can distinguish between different extinction fractions, because the maximized probability of the data is exactly the same for all $\varepsilon$ ($\varepsilon \leq 1$). For a given clade ($n > 1$), the process with positive net diversification ($r > 0$) is mathematically indistinguishable from the process with zero net diversification ($r = 0$).

This result can be interpreted graphically as follows: for any clade with $n > 1$, an infinite number of birth-death parameterizations—including one with zero net diversification—can predict perfectly the observed clade diversity (Supplementary Fig. 1); moreover, all of these parameterizations have identical probability and complexity. A researcher may choose to compute a positive

net diversification rate for a clade ($r > 0$), but this is an assumption, not a result that is supported by the data. We do not claim that an $r = 0$, constant-rate process provides a good explanation for any observed age-richness data, only that such a scenario cannot be distinguished from the $r > 0$ process on the basis of a single datum.

To perform a comparative study on clade diversification using ARR estimates, researchers typically assume a fixed value of the extinction fraction $\varepsilon$ and compute estimates of $r$ conditional on the assumed value. Equivalently, researchers could assume that all clades have identical net diversification rates but vary in their extinction fraction. In Supplementary Note 1, we show that the maximum likelihood estimate of the extinction fraction is given by

$$\hat{\varepsilon} = \frac{n - e^{rt}}{n - 1} \tag{3}$$

and we prove that the probability of a given age-richness datum at the maximum is also given by Eq. (2). Thus, for a given cross-clade comparison of ARR estimates, researchers could "invert" the analysis by arbitrarily assigning a fixed nonzero value of $r$ to all clades and computing the corresponding maximum likelihood estimates of $\varepsilon$ for each clade. This scenario (clades differ only in $\varepsilon$, not $r$) has exactly the same probability and complexity as one where clades differ only in $r$. However, the biological interpretation is profoundly different from traditional ARR studies, because the variation in species richness emerges from a single (invariant) net diversification rate across clades. These mathematical results pertain only to the use of ARR point estimates for single clades and not to the use of data from multiple clades to estimate a single net diversification rate or extinction fraction[9,23–25].

**Empirical analysis**. We analyzed 15 sampling-standardized paleontological time series of diversity through time, spanning a variety of taxonomic groups. Most paleontological estimates of species richness can be interpreted as being proportional to, not equal to, true richness (e.g., shareholder quorum subsampling; SQS[2,26,27]). Therefore, we estimated total diversity through time using scaling factors applied to paleontological richness estimates after compiling information on species-level richness for living and extinct clades ("Methods"). We obtained estimates of clade age from a variety of paleontological and molecular phylogenetic sources ("Methods" and Supplementary Note 3; Supplementary Table 1). The 15 focal clades span a variety of timescales and diversity trajectories, including clades that are extinct (e.g., graptoloids, trilobites). Others are representatives of Sepkoski's "modern fauna", such as bivalves and gastropods[28], that are thought to have increased substantially in diversity towards the recent (Supplementary Fig. 2). The richness trajectories of these latter clades should conform most closely to the assumption of positive net diversification that underlie the ARR estimator.

We computed ARR estimates of net diversification rate for each sampled timepoint in the fossil diversity trajectories, under the most commonly assumed extinction fractions of $\varepsilon = 0.5$ and $\varepsilon = 0.9$. The logic underlying these rate calculations is illustrated in Fig. 1. For each clade, the ARR estimates decrease with the timescale of measurement (Fig. 2). The tendency for rates to covary negatively with clade age is apparent regardless of the shape of the underlying diversity trajectory: extinct, extant, and rapidly-radiating clades all show extreme declines in ARR estimators through time. The patterns shown in Fig. 2 are nearly identical to those that we would obtain if species richness through time is sampled from a uniform distribution with no underlying biological process (Supplementary Fig. 3). In Supplementary Note 2, we explore the possibility that rates can be explained by the so-called "Push of the Past" (POTP), which can lead to

overestimation of true diversification rates early in a clade's history as a result of survivor bias[25,29,30]. POTP alone is unlikely to cause the time-scaling that we report (Supplementary Fig. 4), and furthermore is deeply problematic for ARR studies, because POTP will yield faster rate estimates for younger clades in the present day, even when true rates are invariant. We also demonstrate that ARR rates for subclades within a major clade can be highly unstable: that is, the ARR rates for a given set of subclades (e.g., trilobite orders) at a given timepoint need not have any correlation with rates computed for the same set of subclades at a different point in time (Supplementary Note 2; Supplementary Fig. 5).

These results reject the hypothesis that biological or biogeographic attributes of clades are the primary determinant of ARR. In fact, the strongest signal in time series of ARR values is simply the time elapsed since clade origin, and this time-correlated variation is large. Across the 15 focal clades, the ARR estimates drop by more than a full order of magnitude on average between the first (oldest) and last (most recent) timepoints. This pattern of decline is similar to an exponential decay process and dwarfs any features associated with the diversity trajectories of groups. This result is consistent across extant and extinct clades and does not depend appreciably on the assumed relative extinction rate (Supplementary Table 2). The reason for this phenomenon is simple: no matter when in the history of a clade we observe its (fossil record) diversity, the ARR estimators predict that diversity should be increasing rapidly (Fig. 1a–c; Supplementary Fig. 6). Because diversity is not rising exponentially for most clades, calculation of ARR rates over progressively longer timescales yields estimates that scale negatively with the duration over which they are computed.

To test whether ARR estimates are predictive, we asked whether rate estimates for a given point in time can predict species richness at some other timepoint in the same fossil diversity series (Fig. 1d–f). Given a clade with a known diversity history and age, we consider a focal timepoint and its associated diversity value. Using this age-richness datum, we compute the ARR estimate for the clade. We then test whether this ARR estimate predicts the species diversity of the clade at some other time, for which an independent estimate of clade diversity is available from the fossil record. We computed the pairwise prediction error for all 25,740 pairs of timepoints across the 15 datasets and analyzed the results as a function of the lagged temporal difference between timepoints.

The ARR estimator fails dramatically at predicting future diversity (positive lags; Fig. 3). For most diversity series and, as expected when a non-exponential process is mis-specified as being exponential, prediction errors are large. In our analyses, they exceed the total number of described species on Earth (Fig. 3, dotted line) at lags of 100–200 million years. Results for negative lags are shown in Supplementary Fig. 7 and show a general tendency towards underprediction of richness at negative lags (Fig. 1e). Note that, under the ARR scenario, theoretical maximum error for negative lags is bounded by the ARR assumptions that expected clade diversity in the past cannot exceed the clade's present-day richness, as illustrated by the predicted curves in Fig. 1c. Surprisingly, ARR estimators even perform poorly for clades that have undergone rapid diversity increases through time, such as bivalves and gastropods (Supplementary Fig. 2; Fig. 2, gray polygons). Although the diversity of these groups has increased through time, it has increased far less than we would expect under the geometric increase scenario assumed by the ARR estimator (Fig. 1, Supplementary Figs. 1, 6).

Gastropods, for example, have increased approximately tenfold in diversity since the Early Cretaceous[2,28], with a present-day

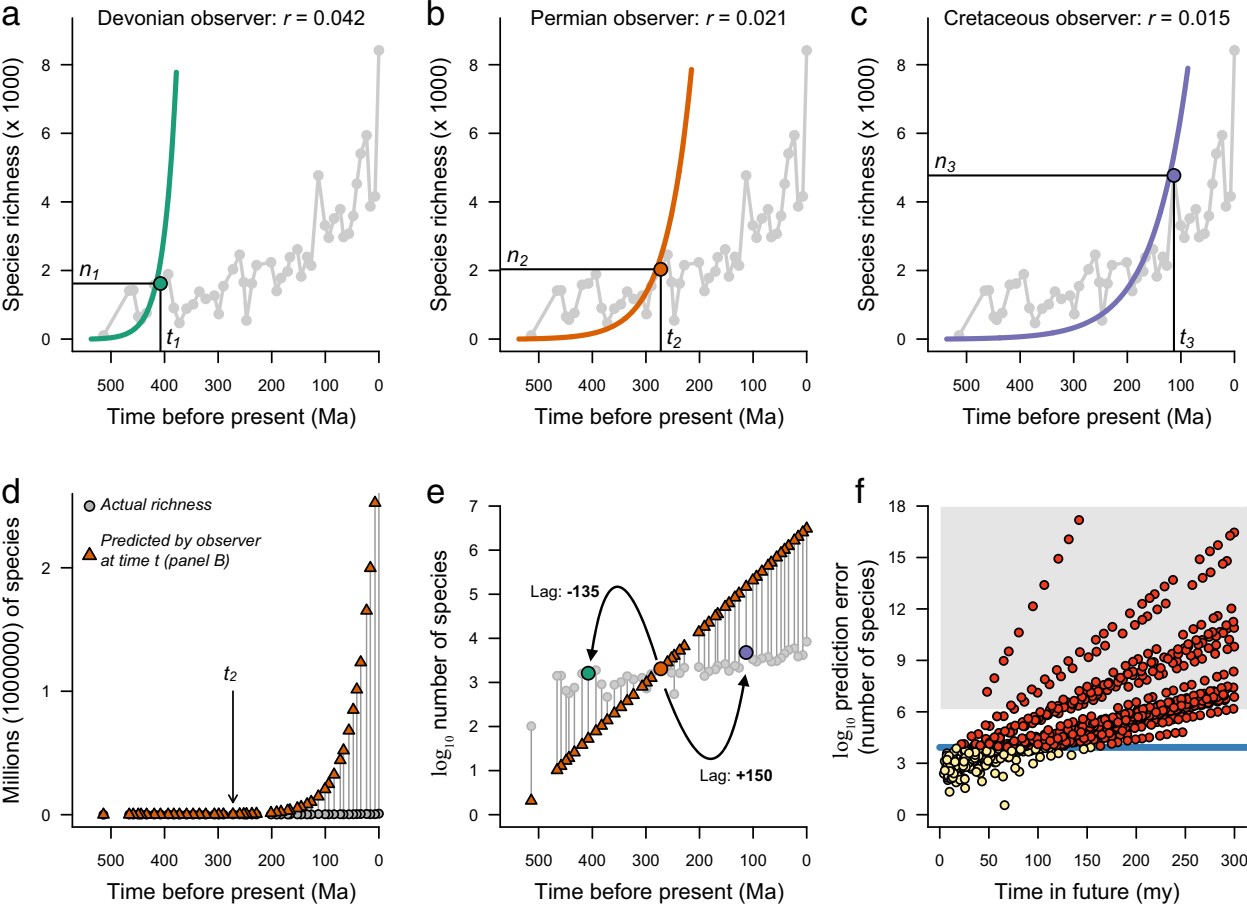

**Fig. 1 Validation of ARR rates using paleontological time-series data.** ARR estimates of net diversification rate for a single clade from the perspective of an observer at three different reference times in the Phanerozoic: **a** Devonian, **b** Permian, and **c** Cretaceous. At any given point in time, the ARR estimate is computed using only the current standing diversity and clade age (e.g., $n_1$ and $t_1$ for the Devonian rate in **a**). Gray curve shows the true diversity trajectory for bivalves, which is either unobserved or otherwise unused by the observer in each reference frame. Smooth exponential curves give the expected species richness through time under the ARR estimate and always predict perfectly the observed diversity at the focal point ($t_1$, $t_2$, or $t_3$). Each curve predicts a rapid rise in species richness after the reference point. **d** Actual versus predicted species richness associated with the Permian rate ($r_2$ at time $t_2$), with $y$-axis scale in units of $10^6$ species. Vertical lines between points give the prediction error relative to the true diversity series (gray circles). After the focal time $t_2$, species richness is predicted to increase explosively under the ARR rate. **e** Same as **d**, but with species richness plotted on $\log_{10}$ scale, illustrating that the ARR estimator $r_2$ consistently underestimates species richness at early points in clade history. Prediction error is a function of the temporal difference or "lag time" between each pair of points; here we show how the lag time is computed for timepoints $t_1$ and $t_3$ relative to $t_2$. **f** Forward-time (positive lag) prediction error for all 1081 pairs of timepoints for bivalves, on $\log_{10}$ scale. Red points indicate pairs of timepoints with prediction errors greater than the maximum number of species in the clade; gray polygon denotes prediction errors that exceed the number of described species on Earth. For nearly any point in the bivalve diversity trajectory, the computed ARR estimate fails spectacularly at predicting future species richness. For this example, rates were estimated with Eq. (1) under a relative extinction rate of $\varepsilon = 0.9$, assuming the origin of bivalves at 530 Ma.

marine diversity of roughly 37,000 species. Given a Cambrian stem age for Gastropoda, an Early Cretaceous observer would compute ARR rates ranging from $r = 0.015$ to $r = 0.019$ under the most commonly-assumed relative extinction fractions from previous ARR studies. These values predict astronomical numbers of gastropod species in the present day (12,000,000 to 110,000,000 species), a prediction error that far exceeds the number of described species on Earth. The ARR estimate performs even worse if we assume that there are additional species to be discovered, because the conversion factors we applied to the SQS richness estimates would have been too low, resulting in systematic underestimates of both the ARR rates and their prediction errors. Once adjusted for the correct level of historical diversity, the ARR projections for future richness will be exponentially (not linearly) greater than any predictions based on diversity undercounts. For example, if the true number of present-day gastropods is actually on the order of 100,000 species,

then the ARR estimates computed for the Early Cretaceous would overpredict present-day richness by many billions ($10^9$) of species.

We next compared ARR performance to null models for species diversity that should have low predictive power for real datasets if an ARR-like process governs the dynamics of species richness through time. In the "constant" model, we assume that the richness at some other time is identical to the richness at the focal time. This model asks: how well does current diversity predict past or future diversity independent of time? We then considered a "zero" model, where we assume that clade richness is due to a birth-death process but where net diversification rates have been zero at all times in the clade's history. For each pair of timepoints, we compared the evidence for the ARR scenario relative to each of the two null models using Akaike weights. We expect the ARR model to fit much better than the constant model, except at lags close to zero, where the models

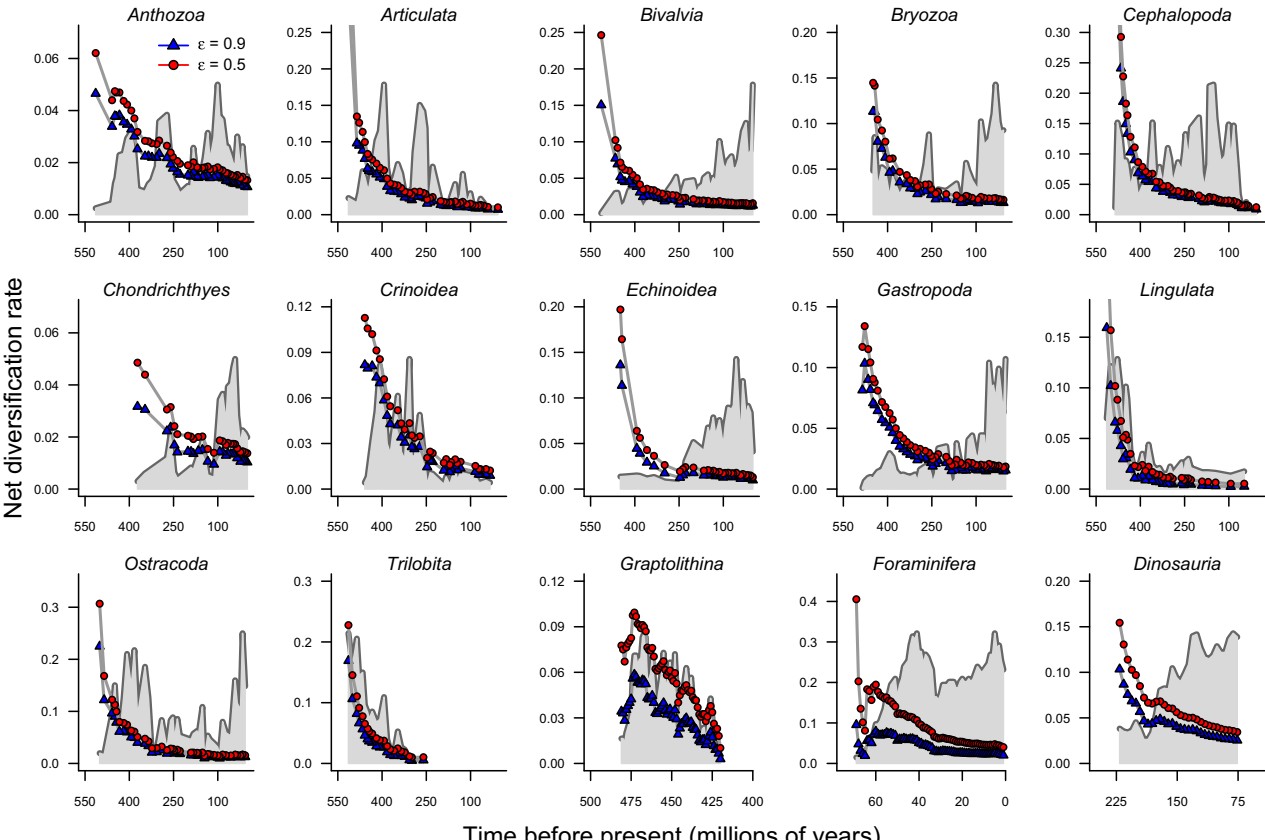

**Fig. 2 Time-scaling of ARR rates.** ARR estimates of net diversification rate decrease sharply through time as a function of the timescale over which they are computed. For each timepoint, rates were computed using an estimate of stem clade age and the standing diversity at the focal time, as in Fig. 1 **a–c** Rates were computed using relative extinction rates of 0.5 (red circles) and 0.9 (blue triangles), which are widely used in neontological studies. Gray polygons show the underlying diversity trend for each clade on a proportional scale (see Supplementary Fig. 2 for actual values). The decay in rates occurs regardless of the form of the underlying diversity trend. Clades that increased in diversity through time (e.g., Bivalvia, Gastropoda, Dinosauria) show the same general pattern as extinct or declining clades. This temporal decay in rates is the expected pattern under null models where species richness fluctuates randomly in time with no underlying biological process (Supplementary Fig. 3).

should have equivalent explanatory power (Supplementary Fig. 6).

Across all datasets, the ARR estimator performs worse than the constant model at predicting species richness (Fig. 4). Near the focal timepoint $t_1$, with lags approaching zero, the constant and ARR scenarios perform equivalently (weight = 0.5). Both the constant and ARR models are able to capture autocorrelation in diversity near the focal timepoint and thus retain predictive ability as the lag approaches zero. However, AIC weights for ARR drop to approximately zero as the absolute lag increases for all datasets. The ARR model is almost never strongly preferred over the constant model: only 1.8% of lagged pairs across all datasets favored the ARR model with weight greater than 0.95. Conversely, 46.5% of timepoints strongly reject (weight < 0.05) the ARR model in favor of the constant model. Virtually identical results are found for the zero ($r = 0$) model, where just 0.2% of timepoints across all datasets strongly favored the ARR model, versus 46.6% strongly favoring the zero model (Supplementary Figs. 9–11). In the Supplementary Information, we include a parallel set of analyses where we assess absolute error in predicted richness for constant and zero models, and we added a "random" model. The random model assumes that species richness is drawn from a uniform distribution with an upper bound set to the maximum diversity ever observed for each clade (we were unable to perform an AIC-based assessment of the random model; see Supplementary Note 2). Across all datasets, the ARR estimator performs worse than nonbiological null models at predicting species richness with respect to absolute error in richness (Supplementary Tables 3, 4).

## Discussion

In the absence of additional information from the fossil record or from time-calibrated phylogenies, ARR "point estimates" should not be used to compare net diversification rates across clades. We have proven that the likelihood of a given age-richness pair is exactly the same under both positive and zero net diversification rates. Therefore, the process indexed by the ARR is not identifiable. Moreover, the fundamental assumption of ARR comparative studies—that the net diversification rate $r$ varies across clades yet the extinction fraction $\varepsilon$ does not—is untenable. We have shown that any ARR dataset is identical in both probability and complexity to an alternative formulation where all clades have the same net diversification rate but differ only in $\varepsilon$.

The two major theoretical issues we describe for ARR (saturated model; non-identifiable process) resulted in predictable pathologies for all paleontological diversity series that we examined. Across a broad range of taxa and timescales, the ARR estimator decays predictably over the timescale of measurement (Fig. 2) and shows virtually no predictive accuracy (Figs. 3, 4, Supplementary Fig. 10). The estimator is outperformed by a simpler metric (current diversity) that discards all information on clade age. Within individual time series, the value of the ARR

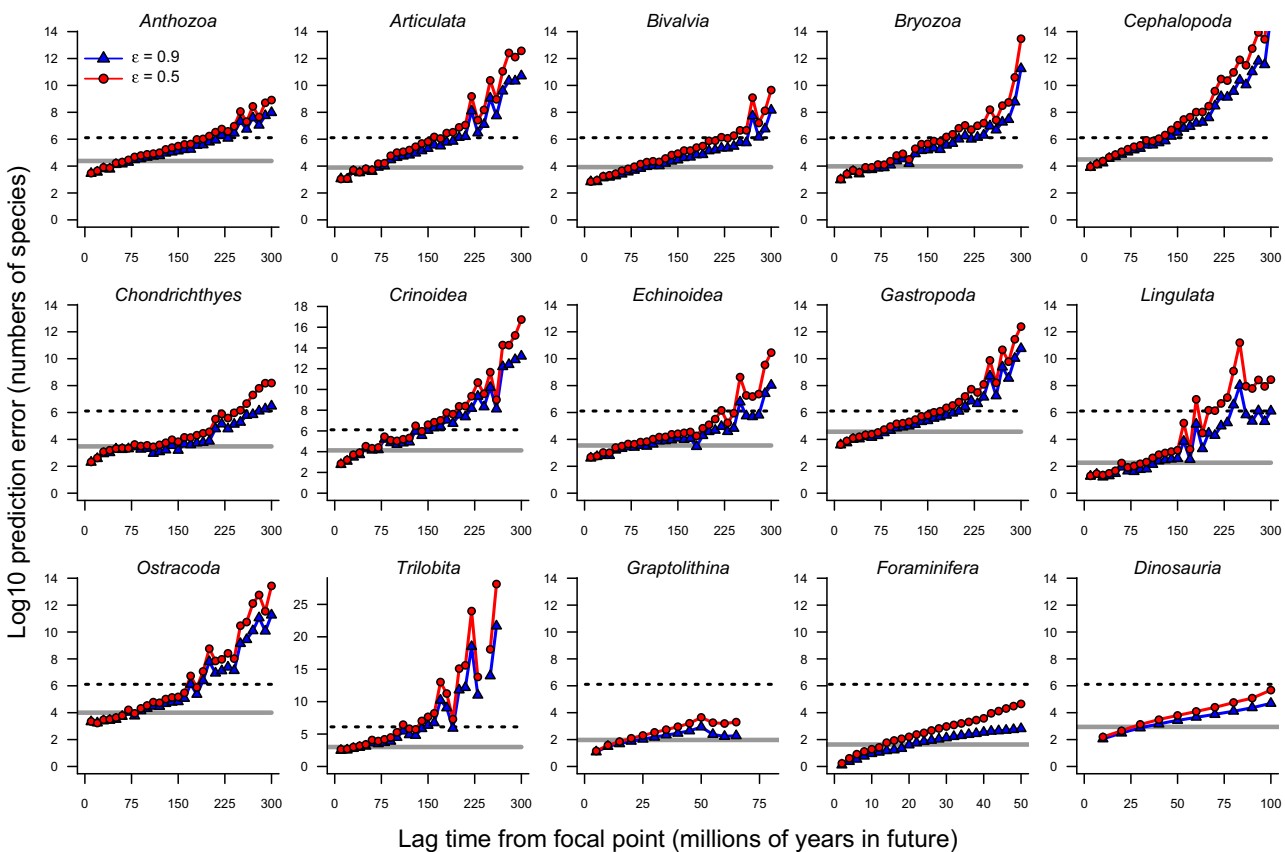

**Fig. 3 Predictive accuracy of ARR estimates of net diversification rate.** Forward-time prediction error (positive lag) for species richness under ARR rate estimates ($\varepsilon$ = 0.5: red circles; $\varepsilon$ = 0.9: blue triangles) for paleontological diversity-through-time series. Solid gray line denotes maximum richness observed for each clade; dotted line is number of described species on Earth. ARR prediction errors increase rapidly as a function of lag time, even for clades that appear to be undergoing rapid increases in diversity through time (e.g., Bivalvia, Gastropoda, Dinosauria; Supplementary Fig. 1). Points show median absolute prediction error in species richness for all pairs of timepoints that fall within a particular lag bin (e.g., +15 to +30 my). ARR estimates of net diversification rate have virtually no ability to predict future diversity for any paleontological time series.

estimator is largely determined by the age of the clade (Fig. 2). This result is consistent with previous studies that have documented a tendency for evolutionary rates to covary negatively with the duration over which they are measured[31–33]. Moreover, paleobiologists have long been aware that exponential growth generally provides a poor approximation to clade dynamics in the fossil record[34–39], while also acknowledging that simple models of this kind retain some context-dependent utility[10].

A recent study demonstrated that diversification inferences from time-calibrated phylogenetic trees are unreliable, because large (potentially infinite) sets of diversification scenarios can have identical probability for a given dataset[40]. For the age-richness datasets considered by our study, the problem is even more severe, because any given age-richness datum is equiprobable under an infinite set of equally-complex parameterizations and because there are no residual degrees of freedom with which to assess model adequacy.

Some researchers nonetheless claim that the negative correlation between ARR estimates and clade age is not problematic, hypothesizing that older clades have biological attributes that cause them to have slower net diversification rates[41,42]. Our results reject this hypothesis, by showing that rates decline monotonically within the diversity time series of individual clades. ARR declines sharply through time even for those clades that appear to have undergone rapid increases in diversity towards the present, such as bivalves and gastropods[2,28]. The order-of-magnitude decline in ARR through time for most clades

(Fig. 2; Supplementary Table 2) is much greater than the range of among-clade variation that many ARR studies have sought to explain. For example, a recent ARR comparative study reported a maximum difference of just 0.024 lineages per my separating the fastest- and slowest phylum-level animal clades[43]. Importantly, the details of why rates show time-scaling are largely irrelevant[31]: the fact that it exists in most empirical datasets is inherently problematic for ARR studies, because some or all of the variation in ARR rates among clades may simply reflect differences in clade age and not the action of clade-specific traits.

Figure 5 illustrates six scenarios whereby two clades differ in their present-day ARR estimate but where the differences have no meaningful relationship to biological process. In this example, the lineages comprising red and blue clades are fully identical and exchangeable: one can view the trajectories as independent evolutionary experiments using groups with exactly equivalent organismal and biogeographic traits, for which the only difference is when in time the trajectories were started. For example, an old clade may have experienced a mass extinction that occurred prior to the origin of the young clade. Even if both clades have identical diversification rates whenever they are contemporaneous (Fig. 5a), such a scenario will typically result in lower present-day ARR estimates for the old clade (blue) relative to the young clade (red). The young clade avoids the impact of the mass extinction on its present-day ARR estimate simply because it is young. Under all six scenarios (Fig. 5), analyses of ARR rate estimates with respect to differences in clade traits or

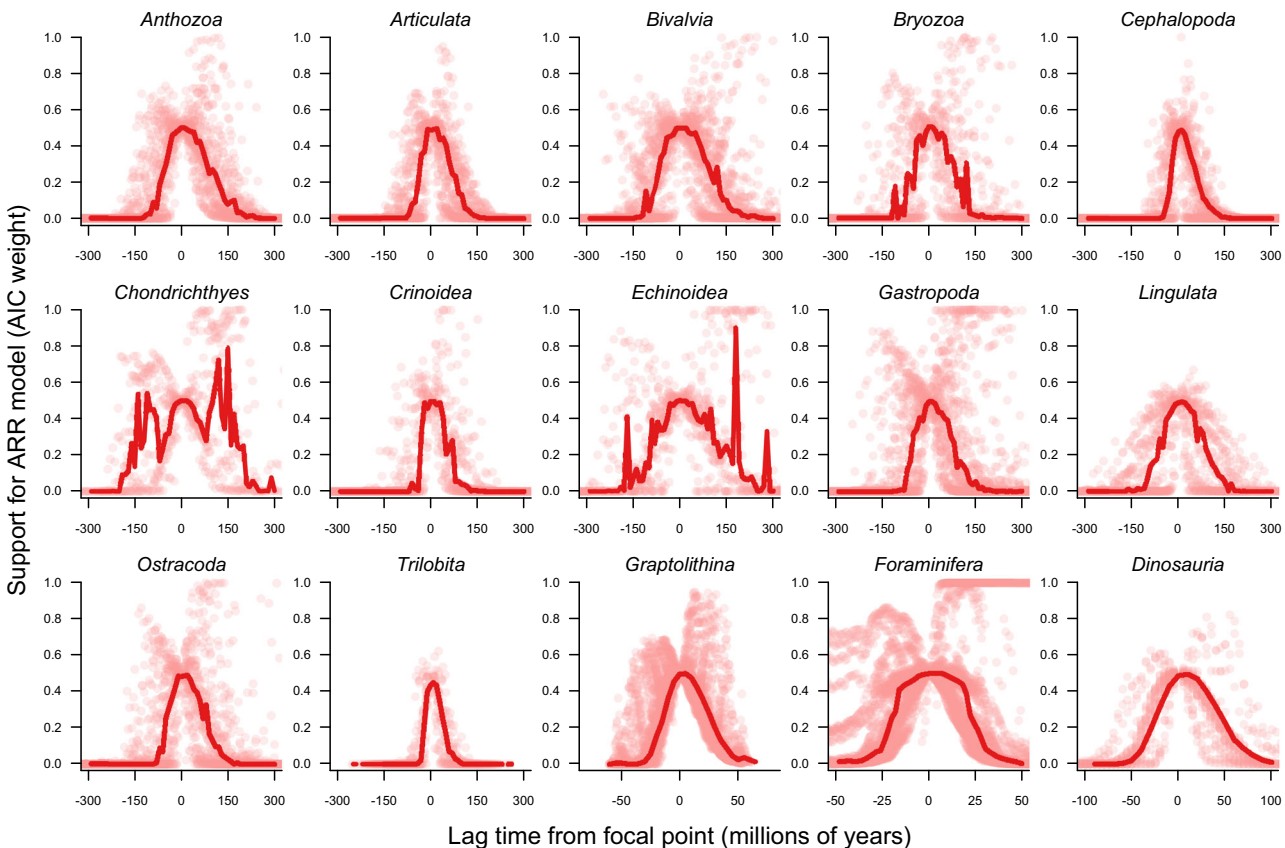

**Fig. 4 Probability (AIC weight) of ARR model relative to a nonbiological "constant" model as a function of temporal lag.** Weights are shown for all pairwise combinations of timepoints for each dataset; red line denotes median weight of ARR model for binned lag times. A value of 0.5 indicates that ARR and constant models have equivalent probability; values approaching 1 imply superior performance of the ARR model. For both positive lags (forwards-in-time) and negative lags (backwards-in-time), the ARR model is outperformed by the constant model. Across all pairs of timepoints ($n = 25740$), the ARR model was strongly favored (weight $> 0.95$) for only 1.8% of comparisons. In contrast, the constant model was strongly favored in 45.5% of comparisons. Beyond a certain point in both directions, the ARR model has minimal explanatory power for any dataset. The number of parameters in ARR and constant models is identical ($n = 1$), and identical weight implies identical probability. Corresponding analyses for other models are shown in Supplementary Figs. 9–11.

biogeography would be deeply misleading, because the apparent rate differences are not caused by any properties of clades themselves. Rather, apparent rate differences reflect historical contingencies (Fig. 5a–c), statistical "survivorship bias" (Fig. 5d), or temporal offset of otherwise identical diversification trajectories (Fig. 5e, f).

The ARR estimator has enjoyed wide use in evolution and ecology because the data to compute the estimates are readily available, and because of a general perception that simple methods can be relatively robust to violation of their assumptions. The problem, in this case, is not the simplicity of the inference model, but the fact that the inference model is saturated: its parameters are estimated from a single data point (Fig. 1). Because the inference model is saturated, it can perfectly explain all possible age and richness values (Supplementary Fig. 1), and there is no remaining information in the data that can be used to test the validity of the resulting estimates. No matter what process generates the underlying data – random noise, measurement error, or other biological processes – the researcher will always obtain ARR estimates that are perfectly consistent with those data (as shown in Fig. 1a–c).

ARR estimators themselves retain considerable utility in some contexts, particularly for parameterizing null hypotheses of clade diversification[11,29,44]. However, the increasing availability of time-calibrated phylogenies for higher taxonomic groups, and

resulting ease of estimating clade ages, has led to the widespread use of ARR in cross-clade comparative studies. Many studies have applied ARR estimators on timescales that far exceed those considered by our study, including comparisons among kingdom and phylum-level clades that vary in age by hundreds of millions[43] to billions of years[45]. At these temporal scales, it is likely that the ARR is nothing more than a number with the property of being computable from two other numbers.

Our results demonstrate that ARR point estimates should not be used in cross-clade comparative studies, unless external validation can be provided by referencing additional paleontological or phylogenetic data. Minimally, researchers who wish to use the ARR estimator for comparing rates across higher taxa should apply them in the context of an unsaturated statistical model that draws information from multiple clades in a model selection framework[9,23], or provide independent checks on the rate estimates using analyses of species-level phylogenies. However, our results further suggest that we may need to abandon the notion of a biologically-meaningful "net diversification rate" that can be used simplistically to compare clades with histories spanning tens to hundreds of millions of years. Using such indices in downstream inference, with no consideration of how and why those rates may fail to describe the dynamics of clade diversity, is likely to yield spurious conclusions about the causes of species richness in time and space.

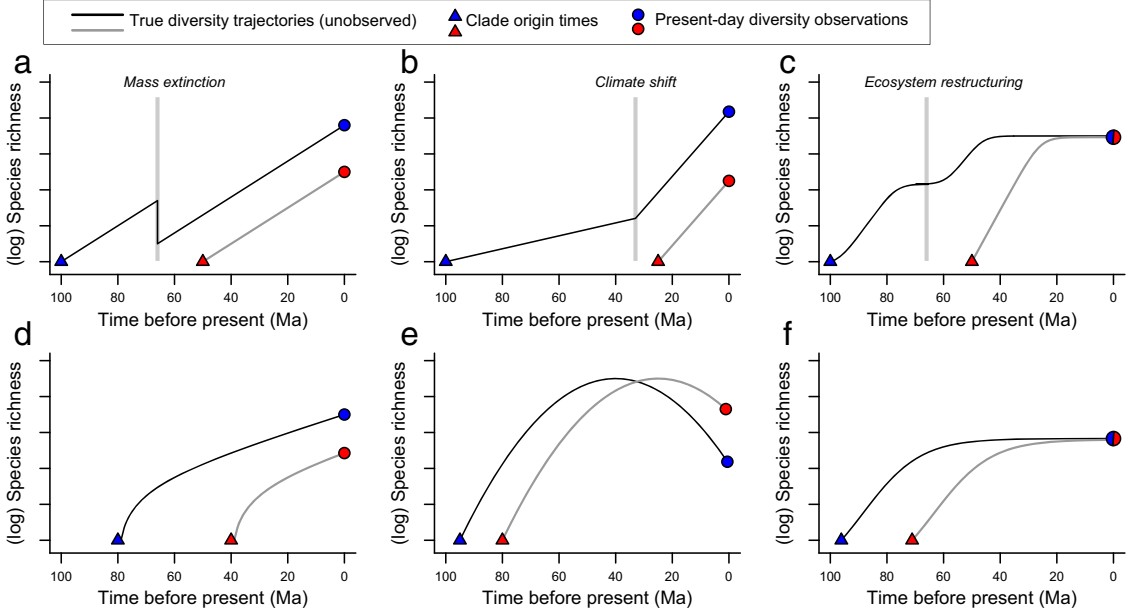

**Fig. 5 Example diversity trajectories where ARR rates will be positively misleading.** Differences in ARR estimates readily emerge for clades comprised of species with fully identical biological properties, such that any association between rates and traits is positively misleading. In each scenario, the estimated present-day rate for the red clade is faster than the corresponding blue clade rate, and this apparent rate variation is due solely to the greater age of the blue clade. Triangles denote clade origin, circles denote the present-day, and lines indicate true (unobserved) diversity trajectories. **a** Clades have identical true diversification rates, but the older clade experienced a mass extinction at the KPg boundary, before the origin of the red clade. **b** Clades share a common environment-dependent net diversification rate and diversify at exactly the same rate whenever they are contemporaneous. **c** Clades have an identical time-varying carrying capacity. **d** Clades have identical diversity trajectories, offset in time, showing "push of the past"[29]. **e** Clades have identical diversity trajectories, offset in time, showing symmetric waxing and waning of each clade[57,58]. **f** Clades have identical diversity trajectories, offset in time, but are regulated at the same level by diversity-dependent factors.

## Methods

**Mathematical analysis**. In Supplementary Note 1, we derive the maximum likelihood estimator (MLE) of the net diversification rate of a clade for the process beginning with a single ancestral lineage (stem clade) under a positive ($r > 0$) diversification process. We show that the MLE is identical to the "method-of-moments" estimator[11]. We then derive the expression for the maximized log-likelihood of the data as a function of the relative extinction rate, which gives equation [2]. We repeat the exercise for the balanced ($r = 0$) diversification process and show that the MLE of the speciation rate is given by $\lambda = (n-1)\,t^{-1}$ where $n$ and $t$ are the species richness and stem age of the focal clade. On substitution into the likelihood of the survival-conditioned and balanced ($r = 0$) process, we show that the probability of the data at the maximum is given by equation [2]. We then derive the MLE of the relative extinction fraction as a function of any arbitrarily chosen $r$ (equation [3]), and we show that the probability of the data at the maximum is also identical to equation [2].

**Fossil diversity data and clade age**. We assembled diversity trajectories for fifteen clades from previous studies (Supplementary Table 1), predominantly using subsampled diversity estimates for marine animals[2,26]. For nominally species-level analyses of groups that have been proposed to show relatively little provincialism (macroperforate foraminifera[46], graptoloids[47,48]), we made no further adjustments to the data. For extant clades of marine invertebrates, we rescaled each sampling-standardized diversity series using an estimate of the clade's present-day species richness, with the assumption that SQS estimates are proportional to instantaneous species richness levels within a particular time bin[2,27]. The ratio between present-day diversity and the SQS estimate for the most recent Cenozoic time bin was used to rescale the time series into an estimate of the species-level diversity curve. Estimates of current marine species richness for each clade were taken from the Ocean Biogeographic Information System, a comprehensive database of taxonomic information for marine organisms[49]. Maximal dinosaur standing diversity was based on[50] following the reasoning outlined by[51]. Standing diversity of trilobites during the Ordovician is generally assumed to exceed 1000 species, and this number is likely to be very conservative in light of strong geographic sampling biases[52]; we rescaled SQS trilobite richness by assuming that the peak Ordovician diversity for a single time slice was 1000 species. Marine animal clades from Alroy[2] were analyzed in their original time bins, of roughly 10 million years (my) duration, with updated numerical ages based on revisions to the geological timescale[53]. Estimated standing richness for dinosaurs and foraminiferans were lineage counts taken at equally-spaced timepoints from time-calibrated phylogenies for each taxon ([46]; "mbl" phylogeny from[54]). Time slices for dinosaurs, foraminiferans, and

graptiloids were 5, 1, and 1 million years, respectively. Due to qualitative sampling differences between avian and non-avian Dinosauria, we excluded a single descendant subclade (Aves) from diversity estimates for this clade. We identified stem ages for each clade in our dataset by systematically reviewing both paleontological and molecular phylogenetic studies on the early history of each group (Supplementary Note 2). One potential bias for SQS and other proportional diversity estimators involves secular changes in the evenness of taxonomic assemblages through time. For this bias to impact our analyses, evenness should progressively decrease through time, such that true diversity is increasingly underestimated towards the present. However, there is little evidence for such trends, and evenness for marine invertebrates generally appears to have weakly increased or plateaued across much of the Phanerozoic[55].

**Time-dependency of ARR estimates**. We computed ARR estimates for timepoints from each fossil series assuming $\varepsilon = 0.5$ and $\varepsilon = 0.9$. To assess the expected decline in ARR estimates under the POTP[29], we simulated diversity trajectories for each clade conditional on the present-day diversity, then recomputed the ARR estimates for each sampled timepoint using the realized diversity value (Supplementary Note 2). To generate the expected pattern under random noise, we simulated random diversity values for each timepoint and clade by drawing from an integer-valued uniform (1, $N_{max}$) distribution, where $N_{max}$ is the greatest species richness observed in any time interval for the focal clade.

**Analysis of prediction error**. For each timepoint from the fossil series for which a diversity estimate was available, we used the stem clade ARR estimator to predict diversity at all other past and future timepoints from the paleontological series (Fig. 1). We performed all analyses using ARR estimators with $\varepsilon = 0.5$ and $\varepsilon = 0.9$. For each prediction pair ($t_1$, $t_2$), we first computed the estimated ARR estimate $r_1$ at time $t_1$; this rate was then used to compute the expected number of species at time $t_2$, conditioned on clade survival to that time[10]. Mathematical details for the prediction model are provided in Supplementary Note 1. Analyses of prediction error and associated simulations were performed in the R computing environment (R version 3.7.3).

**Model comparisons**. For each pair of timepoints ($t_1$, $t_2$) from a single diversity series, we computed the probability of the observed data $n_2$ at time $t_2$ using $r_1$, the ARR estimate from time $t_1$. Conditional on clade survival to the observation time, the probability is given by $P(n_2, t_2 \mid r_1, \varepsilon) = (1-\beta)\,\beta^{(n_2-1)}$, which is a geometric distribution of species richness with parameter $1 - \beta$ (Supplementary

Note 1). To compute the probability of the data under the constant model, we assumed that richness at time $t_2$ $(=n_2)$ is also drawn from a geometric distribution, but with mean equal to the richness at time $t_1$ $(=n_1)$. This model is intended to be nonbiological, but can also be derived as a formal diversity-dependent model where carrying capacities among clades follow a geometric distribution[56]. We computed the AIC weight of the ARR model for each dataset and temporal lag class. Because both the ARR and constant models have the same number of parameters, the "weight" of the ARR model is $P_{ARR}/(P_{ARR} + P_{CONST})$, where $P_{ARR}$ and $P_{CONST}$ are the probabilities of the observed richness $n_2$ under the ARR and constant models, respectively. We repeated this exercise for the $r = 0$ ("zero") model, conditioned on clade survival to the present. Importantly, this latter model predicts a linear increase in richness with time after conditioning on clade survival to the observation time (Supplementary Note 1). One can also think about the "zero" model as a "linear" model, to contrast with the exponential growth scenario specified by the ARR model. Analyses of absolute prediction errors (non-probabilistic) are described in Supplementary Note 2.

**Reporting summary**. Further information on research design is available in the Nature Research Reporting Summary linked to this article.

## Data availability

All data analyzed in this study are publicly available. Data for macroperforate foraminifera are available at https://onlinelibrary.wiley.com/doi/full/10.1111/j.1469-185X.2011.00178.x. Dinosaur and graptoloid data were downloaded from the Dryad digital repository at https://datadryad.org/stash/dataset/doi:10.5061/dryad.gr1qp and https://datadryad.org/stash/dataset/doi:10.5061/dryad.fq7h2, respectively. Raw occurrence data used to generate the diversity curves for all other clades are available through the Paleobiology Database (https://paleobiodb.org). Compiled datasets, including fossil diversity time series and associated clade ages, are available as part of the data package that accompanies this article through the Dryad digital data repository (https://doi.org/10.5061/dryad.qz612jmfb).

## Code availability

Computer code to recreate all analyses and figures from this article are available through the Dryad digital data repository (https://doi.org/10.5061/dryad.qz612jmfb).

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

## Acknowledgements

This work was supported in part by a fellowship from the David and Lucile Packard Foundation (D.L.R.).

## Author contributions

D.L.R. and R.B.J.B. conceived and designed the study, compiled data, analyzed data, and wrote the manuscript.

## Competing interests

The authors declare no competing interests.
