## [Peer Review File · Nature Communications]

Reviewers' Comments:

Reviewer #1:

Remarks to the Author:

This is a rare review where I have no major comments on this paper. Having gone through your manuscript and supplement in detail, I am convinced by all of the arguments (although I suspected this was the case for some time). The text is clearly written and lays out the fallacies of this approach in a way that even non-specialists can understand. I also really liked the empirical analysis using the paleo data; while I had intuited the non-identifiability of the ARR estimator, I hadn't previously thought of testing this using fossil time series. This adds a lot to the theoretical treatment. I enthusiastically recommend it for publication, essentially as is.

Indeed, the one point where I would like to challenge you is that perhaps you do not come down strongly enough. You write starting on line 368 that researchers might want to combine information from different clades in a model selection framework. But this is fundamentally at odds with how people want to use this estimator -- that is, in order to combine information from different clades, you have to assume that diversification rates are either identical or come from a common distribution (and the best one could do would be to estimate the hyperparameters of this distribution). That is this would apply the ARR estimator to look for similarity in macroevolutionary dynamics not to parse out variation. Related to this, what would the point be of checking this against estimates from molecular phylogenies. We now understand the full scope of time-variable diversification models that can explain a given dataset equally well (following my paper with Louca in 2020) and while creative priors are probably needed to distinguish between these, I don't think you get anything else by looking at the ARR estimate.

Matt Pennell

Reviewer #2:

Remarks to the Author:

The question of whether net diversification rates are continuous & exponential rather than limited in some manner is an issue of both theoretical and methodological concern for evolutionary biologists. Rabosky & Benson's manuscript touches on both issues by examining how a metric commonly used to assess typical rates of diversification in extant clades (Magallon & Sanderson's age-richness-rate or ARR) predicts patterns in the fossil record. Rabosky & Benson show quite convincingly that the ARR metric badly fails to predict richness patterns in the fossil record given net-diversification rates predicting modern clade richness values. Moreover, because the authors use sampling standardized estimates of clade richnesses over the Phanerozoic, we cannot dismiss this as some bizarre sampling artifact; moreover, given that the fossil record routinely reveals that there are too many fossil species given ARRs predicting "future" richness (either modern or among younger fossils), the results are completely opposite the expectations of exponential diversification + poor fossil records.

This manuscript is really well-written and makes its case very clearly & cogently. If anything, then it might make its case too well: one almost feels like ARR is screaming "mercy!" by the end! Moreover, as I do not think that Magallón & Sanderson intended for their metric to be used so cavalierly, it almost seems unfair to them; indeed, it might be worth noting that the limited cases where this metric might be useful might include angiosperms as they have been diversifying like mad since the Cretaceous.

I also can think of two special-pleading arguments that could explain these results, both of which are "half-baked" in the sense that thinking them through completely pretty much disproves them. One, we could get these results if there are dozens of mass extinctions that basically undo all of the diversification over 10's of millions of years, but yet still preserve enough phylogenetic diversity that

the divergence times used herein are correct. However, that requires a crazy crash boom system that makes any "general" net-diversification rate pretty meaningless: and thus proves the overall point. (And, of course, it flies in the face of evidence from the fossil record) Another is that occupancy distributions have shifted through time so that SQS has an easier time sampling high richness early in histories than later, and thus inflating the apparent richnesses of early groups. This would be completely convoluted and require contradictory patterns within individual clades in order to get the early -> middle -> late comparisons. (Also, there is no general trend in evenness of occupancy patterns over time at either the species-level or genus-level.) These are so out-there that they might seem like more like strawmen than anything else and thus probably are not worth mentioning.

I do think that the authors sell the theoretical & methodological implications of their study a little short. One major theoretical implication is that simple continuous exponential models of diversification don't work. Paleontologists (which includes the 2nd author of this manuscript) have been saying this for years and pushing a variety of richness-dependent diversification models instead. It might be worth making note of that. There is a major methodological implication of these results for Bayesian phylogenetic analyses that use birth+death (+ sampling) to generate prior probabilities for individual topologies when contrasting two different proposed evolutionary histories (e.g., Heath et al. 2014 PNAS 111:E2957). Under the simplest models, these are treated as constant through time: but a necessary corollary of these results is that they cannot be. To this end, "skyline" models in which birth & death (origination & extinction) vary through time (Stadler et al. 2013 PNAS 110:228, Stadler & Smrckova 2016 Biol. Lett. 12:20160273, Warnock et al. 2020 Paleobiol. 46:137) are almost requisite for generating appropriate prior-probabilities for phylogenies. Whether the authors want to suggest that this might be a better way to do what ARR tries to do is up to them!

I have a small number of particular comments.

Lines 167 – 169: "These mathematical results pertain only to the use of ARR point estimates for single clades 12-22 and not to the use of data from multiple clades to estimate a single net diversification rate or extinction fraction 9,23-25."

Aren't the latter studies cases in which people contrasted ARR among sister clades or within & among nested subclades? I think that it would be better to state that the results do not pertain to the interpretation that two sister-clades with very different ARR had different diversification dynamics: that conclusion still is safe. Still, it might be worth noting that how the diversification dynamics might be more complicated than $r_1 > r_2$ throughout both sister-clades' histories.

Lines 172-174: "Most paleontological estimates of species richness can be interpreted as being proportional to, not equal to, true richness (e.g., shareholder quorum subsampling; SQS 2,26,27)."

It is worth noting that this introduces a conservative bias when using modern richness because the SQS richness is an underestimate of the true richness; thus, if SQS is too high, then the true richness is too high.

Lines 314-316: "Some researchers nonetheless claim that the negative correlation between ARR estimates and clade age is not problematic, hypothesizing that older clades have biological attributes that cause them to have slower net diversification rates 41,42. "

It might be worth noting that this flies in the face of fossil data, which show that "older" clades (e.g., trilobites and ammonites) tend to be much more volatile than "younger" clades (e.g., bivalves and gastropods), in that the former show much greater bursts and crashes. Now, the net-diversification might not be different, and some models suggest that higher volatility should coincide with lower maximum richness under logistic models (Walker & Valentine 1984 Am. Nat. 124:887); however, logistic models are (by their basic nature) obviously offering a very different explanation for differences in ARR among clades.

Lines 356-357: "ARR estimators themselves retain considerable utility in some contexts, particularly for parameterizing null hypotheses of clade diversification11,29,44."

I would include contrasting sister clades here. I might also mention those rare cases of "young" clades that are still diversifying.

Figure 1. I just want to state that I think that this figure (busy as it is) is really effective.

Figure 2. This figure also is really effective. I appreciate that the remaining figures probably are necessary to really "prove" the case: but the authors had me with Figures 1 & 2.

Peter Wagner.

Heath, T. A., et al. 2014. The fossilized birth–death process for coherent calibration of divergence–time estimates. *PNAS* 111:E2957–E2966. (10.1073/pnas.1319091111)

Stadler, T., et al. 2013. Birth–death skyline plot reveals temporal changes of epidemic spread in HIV and hepatitis C virus (HCV). *PNAS* 110:228–233. (10.1073/pnas.1207965110)

Stadler, T. and J. Smrckova. 2016. Estimating shifts in diversification rates based on higher-level phylogenies. *Biol. Lett.* 12:20160273. (10.1098/rsbl.2016.0273)

Walker, T. D. and J. W. Valentine. 1984. Equilibrium models of evolutionary species diversity and the number of empty niches. *Am. Nat.* 124:887 – 899.

Warnock, R. C. M., et al. 2020. Assessing the impact of incomplete species sampling on estimates of speciation and extinction rates. *Paleobiol.* 46:137-157. (10.1017/pab.2020.12)

We thank the editor and reviewers for their thoughtful comments on our manuscript. Please see below for point-by-point responses to all comments raised by the referees during the review process.

REVIEWERS' COMMENTS

Reviewer #1 (Remarks to the Author):

This is a rare review where I have no major comments on this paper. Having gone through your manuscript and supplement in detail, I am convinced by all of the arguments (although I suspected this was the case for some time). The text is clearly written and lays out the fallacies of this approach in a way that even non-specialists can understand. I also really liked the empirical analysis using the paleo data; while I had intuited the non-identifiability of the ARR estimator, I hadn't previously thought of testing this using fossil time series. This adds a lot to the theoretical treatment. I enthusiastically recommend it for publication, essentially as is.

Indeed, the one point where I would like to challenge you is that perhaps you do not come down strongly enough. You write starting on line 368 that researchers might want to combine information from different clades in a model selection framework. But this is fundamentally at odds with how people want to use this estimator -- that is, in order to combine information from different clades, you have to assume that diversification rates are either identical or come from a common distribution (and the best one could do would be to estimate the hyperparameters of this distribution). That is this would apply the ARR estimator to look for similarity in macroevolutionary dynamics not to parse out variation. Related to this, what would the point be of checking this against estimates from molecular phylogenies. We now understand the full scope of time-variable diversification models that can explain a given dataset equally well (following my paper with Louca in 2020) and while creative priors are probably needed to distinguish between these, I don't think you get anything else by looking at the ARR estimate.

>>> We agree with the reviewer on these points and would be happy to come down even harder on ARR. That said, the field has really struggled with these issues in the recent past, and for this reason we hesitate to be more critical than we already are. We are fairly blunt in our message already, and we worry slightly that coming down even more forcefully/generally will potentially elicit a reactionary response from some of the readers who most need to consider these points carefully in their own work.

Matt Pennell

Reviewer #2 (Remarks to the Author):

The question of whether net diversification rates are continuous & exponential rather than limited in some manner is an issue of both theoretical and methodological concern for evolutionary biologists. Rabosky & Benson's manuscript touches on both issues by examining

how a metric commonly used to assess typical rates of diversification in extant clades (Magallon & Sanderson's age-richness-rate or ARR) predicts patterns in the fossil record. Rabosky & Benson show quite convincingly that the ARR metric badly fails to predict richness patterns in the fossil record given net-diversification rates predicting modern clade richness values. Moreover, because the authors use sampling standardized estimates of clade richnesses over the Phanerozoic, we cannot dismiss this as some bizarre sampling artifact; moreover, given that the fossil record routinely reveals that there are too many fossil species given ARRs predicting "future" richness (either modern or among younger fossils), the results are completely opposite the expectations of exponential diversification + poor fossil records.

This manuscript is really well-written and makes its case very clearly & cogently. If anything, then it might make its case too well: one almost feels like ARR is screaming "mercy!" by the end! Moreover, as I do not think that Magallón & Sanderson intended for their metric to be used so cavalierly, it almost seems unfair to them; indeed, it might be worth noting that the limited cases where this metric might be useful might include angiosperms as they have been diversifying like mad since the Cretaceous.

>>> In the case of angiosperms, it is true that that they've had a major expansion of species richness, but even here one can imagine that ARR could cause problems. If much of the expansion of angiosperm diversity happened between the mid-Cretaceous and Eocene, then once again we are potentially faced with the problems we raise in our manuscript. In some ways, the patterns (and ARR problems) we observe with the "modern" fauna - the bivalves and gastropods, for example - might be a cautionary tale for assuming that diversity dynamics for any clade undergoing apparently rapid expansion in diversity over timescales of 50 – 150 million years is well-approximated by the ARR rates.

>>> We certainly agree that Magallón and Sanderson probably did not intend for their metric to be used so cavalierly. That said, some of the issues we raise here would also apply to their original 2001 paper: the rates presented there would likely show many of the same pathologies we describe here. At the 40-150 my timescale they were working with, it is notable that they don't see much of a relationship between clade age and species richness (their Figure 4); such a lack of relationship is an indicator that the data likely show strong deviations from the constant-rate birth-death process that they are assuming.

I also can think of two special-pleading arguments that could explain these results, both of which are "half-baked" in the sense that thinking them through completely pretty much disproves them. One, we could get these results if there are dozens of mass extinctions that basically undo all of the diversification over 10's of millions of years, but yet still preserve enough phylogenetic diversity that the divergence times used herein are correct. However, that requires a crazy crash boom system that makes any "general" net-diversification rate pretty meaningless: and thus proves the overall point. (And, of course, it flies in the face of evidence from the fossil record) Another is that occupancy distributions have shifted through time so that SQS has an easier time sampling high richness early in histories than later, and thus inflating the apparent richnesses of early groups. This would be completely convoluted and require contradictory patterns within

individual clades in order to get the early -> middle -> late comparisons. (Also, there is no general trend in evenness of occupancy patterns over time at either the species-level or genus-level.) These are so out-there that they might seem like more like strawmen than anything else and thus probably are not worth mentioning.

I do think that the authors sell the theoretical & methodological implications of their study a little short. One major theoretical implication is that simple continuous exponential models of diversification don't work. Paleontologists (which includes the 2nd author of this manuscript) have been saying this for years and pushing a variety of richness-dependent diversification models instead. It might be worth making note of that. There is a major methodological implication of these results for Bayesian phylogenetic analyses that use birth+death (+ sampling) to generate prior probabilities for individual topologies when contrasting two different proposed evolutionary histories (e.g., Heath et al. 2014 PNAS 111:E2957). Under the simplest models, these are treated as constant through time: but a necessary corollary of these results is that they cannot be. To this end, "skyline" models in which birth & death (origination & extinction) vary through time (Stadler et al. 2013 PNAS 110:228, Stadler & Smrckova 2016 Biol. Lett. 12:20160273, Warnock et al. 2020 Paleobiol. 46:137) are almost requisite for generating appropriate prior-probabilities for phylogenies. Whether the authors want to suggest that this might be a better way to do what ARR tries to do is up to them!

>>> We strongly agree with the reviewer on this point. That said, we would suggest that caution may be warranted in extrapolating from our results to the application of constant-rate birth-death (CRBD) models in generating prior probabilities for Bayesian phylogenetics. In particular: the surviving clade – that which leads to the reconstructed tree of extant lineages only – might still show distributions of branching times that are broadly consistent with a CRBD process, even when the process overall (including non-extinct lineages) is equilibrational or otherwise strongly departs from assumptions of CRBD model. For example, imagine a clade replacement scenario where diversity is constant through time. One clade undergoes attrition and ultimate extinction due to whatever reason and the other clade experiences a diversity increase by preferentially speciating as diversity in the other clade declines (like Rosenzweig & McCord's competitive speciation model). This scenario could give a double-wedge pattern. Importantly, even though diversity might be more or less constant through time, the branching times in the extant clade might still be well-approximated by a CRBD process, so for phylogenetic reconstruction this approach might still work even though the CRBD process has nothing at all to do with actual patterns of species richness in this example.

>>> In any event, we have lots of independent evidence from extant clades that, at least in some cases, something *similar to* a CRBD process is not completely unreasonable for the distribution of speciation times in the reconstructed tree.... but as we show (and consistent with other work, eg Louca & Pennell and many others), those patterns in the reconstructed phylogeny don't necessarily have anything to do with the dynamics of biodiversity. And consequently, branching times in extant phylogenies are often hopeless for actually getting at the causes of species richness itself. For all of these reasons, we hesitate to extend our

critique to issues of tree priors etc, as the topic probably warrants a more extensive treatment than we can provide.

>>>Regarding the suggestion to note that paleontologists have advocated diversity-dependent models of diversification. We have stated in our discussion that ‘paleobiologists have long been aware that exponential growth provides a poor approximation to clade dynamics’. We are reluctant to state which models specifically might describe fossil record diversity series, because this is a contentious area and we worry about advocating specific alternative models as goes beyond the scope of our analyses. However, for record, both authors feel it is likely that equilbrial dynamics of diversification are widespread at the appropriate phylogenetic scale.

I have a small number of particular comments.

Lines 167 – 169: “These mathematical results pertain only to the use of ARR point estimates for single clades¹²⁻²² and not to the use of data from multiple clades to estimate a single net diversification rate or extinction fraction ^{9,23-25}.”

Aren't the latter studies cases in which people contrasted ARR among sister clades or within & among nested subclades? I think that it would be better to state that the results do not pertain to the interpretation that two sister-clades with very different ARR had different diversification dynamics: that conclusion still is safe. Still, it might be worth noting that how the diversification dynamics might be more complicated than $r_1 > r_2$ throughout both sister-clades' histories.

>>> We would suggest caution on inferences using sister clades, for a number of reasons. For example: one could still use crown-clade ARR estimates on each of a pair of sister clades and obtain meaningless estimates for each clade; by meaningless, we mean that the ARR estimate would simply reflect differences in the crown age and might not have anything to do with whatever is controlling richness in the clades.

>>> There is a more insidious conceptual problem as well. A comparison of ARR rates for sister clades has the dangerous side effect of implying (rather, assuming) that the rate difference is the cause of the richness difference. But if the clades do not conform to the CRBD assumptions, then the cause of the richness difference could be many other factors (e.g., different realized carrying capacities). So, the researcher concludes that "clade A is more diverse than clade B because of faster diversification rate", and as such the emphasis is thrown on explanations involving the rate of speciation and/or extinction. But in the equilbrial scenario, the ARR rates would still be meaningless, and net diversification would be zero - it would be the slopes of speciation and extinction rate curves with respect to diversity that determine total richness, so the appropriate causal explanation involves understanding how & why those rates are functions of richness.

>>> In fact, contrasts in ARR rates for sister clades more-or-less returns the difference in richness itself, regardless of what process generates the difference. This can most easily be seen from considering ARR rates for a pair of sister clades under a pure-birth process, where the 2 rates are of the form $\log(N_1)/t$ and $\log(N_2)/t$: the ratio in rates is just the ratio

of log-transformed richness, since t drops out of the equation. As such, applying ARR probably doesn't bring anything to the analysis of sister clades that isn't evident from inspection of the richness difference itself.

>>> So, we agree with the reviewer that the ARR comparison might tell us that the *dynamics are different*, but (as we've seen throughout our field in general) the use of ARR would all too often lead to the (quite possibly incorrect) conclusion that the cause of the richness difference resides in differential net diversification rate, which doesn't follow from that comparison. And in fact, if the clades are at equilibrium, the computed ARR values would have zero predictive utility going forward in time: the "faster" clade should become proportionately more diverse than the "slower" clade if the ARR rates capture something meaningful, and it's easy to see why this wouldn't be the case if richness is equilibrial.

Lines 172-174: "Most paleontological estimates of species richness can be interpreted as being proportional to, not equal to, true richness (e.g., shareholder quorum subsampling; SQS 2,26,27)."

It is worth noting that this introduces a conservative bias when using modern richness because the SQS richness is an underestimate of the true richness; thus, if SQS is too high, then the true richness is too high.

>>> We agree, although we aren't entirely sure that any biases are assured to be conservative as seems that it could depend on several potential factors influencing the SQS estimates. Hence, we prefer to refrain from discussing this issue. Also note that our specific approach is to attempt to re-scale the SQS estimates to inferred 'true' richness by comparing the SQS richness of recent fossil intervals to the present-day richness. This step probably removes that aspect of downward-bias, or at least is intended to do so.

Lines 314-316: "Some researchers nonetheless claim that the negative correlation between ARR estimates and clade age is not problematic, hypothesizing that older clades have biological attributes that cause them to have slower net diversification rates^{41,42}."

It might be worth noting that this flies in the face of fossil data, which show that "older" clades (e.g., trilobites and ammonites) tend to be much more volatile than "younger" clades (e.g., bivalves and gastropods), in that the former show much greater bursts and crashes. Now, the net-diversification might not be different, and some models suggest that higher volatility should coincide with lower maximum richness under logistic models (Walker & Valentine 1984 *Am. Nat.* 124:887); however, logistic models are (by their basic nature) obviously offering a very different explanation for differences in ARR among clades.

>>> We fully agree with this point. However, we feel that the ARR estimators are failing at an even more fundamental level - e.g., a level that doesn't even require bringing in data on the actual dynamics of older clades. Our analyses show that, even ignoring all the clade-specific biology referenced by the reviewer, we can still show that the ARR estimates have minimal predictive utility. As such, our preference is to avoid bringing in more specialized empirical evidence (even if it helps our argument), for the simple reason that many

neontologists will be unfamiliar with this record and it might slightly distract from our main argument.

Lines 356-357: “ARR estimators themselves retain considerable utility in some contexts, particularly for parameterizing null hypotheses of clade diversification 11,29,44.”

I would include contrasting sister clades here. I might also mention those rare cases of “young” clades that are still diversifying.

>>> See above regarding sister clade contrasts. We agree with the point about young clades that are still diversifying, but remain concerned about the fact that the ARR estimates themselves provide no information about whether the focal clades meet the assumptions of the inference model. For that, researchers would presumably need to bring in additional data to robustly defend the use of the metrics.

Figure 1. I just want to state that I think that this figure (busy as it is) is really effective.

Figure 2. This figure also is really effective. I appreciate that the remaining figures probably are necessary to really “prove” the case: but the authors had me with Figures 1 & 2.

>>> Thanks for the comments on the figures - we hope that they are helpful for the audience of nonspecialists who could benefit from appreciating these points more generally.

Peter Wagner.

- Heath, T. A., et al. 2014. The fossilized birth–death process for coherent calibration of divergence–time estimates. *PNAS* 111:E2957–E2966. (10.1073/pnas.1319091111)
- Stadler, T., et al. 2013. Birth–death skyline plot reveals temporal changes of epidemic spread in HIV and hepatitis C virus (HCV). *PNAS* 110:228–233. (10.1073/pnas.1207965110)
- Stadler, T. and J. Smrckova. 2016. Estimating shifts in diversification rates based on higher-level phylogenies. *Biol. Lett.* 12:20160273. (10.1098/rsbl.2016.0273)
- Walker, T. D. and J. W. Valentine. 1984. Equilibrium models of evolutionary species diversity and the number of empty niches. *Am. Nat.* 124:887 – 899.
- Warnock, R. C. M., et al. 2020. Assessing the impact of incomplete species sampling on estimates of speciation and extinction rates. *Paleobiol.* 46:137-157. (10.1017/pab.2020.12)